# Preparation of Bio-Based Foams with a Uniform Pore Structure by Nanocellulose/Nisin/Waterborne-Polyurethane-Stabilized Pickering Emulsion

**DOI:** 10.3390/polym14235159

**Published:** 2022-11-27

**Authors:** Yiqi Chen, Yujie Duan, Han Zhao, Kelan Liu, Yiqing Liu, Min Wu, Peng Lu

**Affiliations:** College of Light Industry and Food Engineering, Guangxi University, Nanning 530004, China

**Keywords:** TOCNC/Nisin/WPU, Pickering emulsion, biomaterials, porous foam, thermal insulation

## Abstract

Bio-based porous materials can reduce energy consumption and environmental impact, and they have a possible application as packaging materials. In this study, a bio-based porous foam was prepared by using a Pickering emulsion as a template. Nisin and waterborne polyurethane (WPU) were used for physical modification of 2,2,6,6-tetramethyl piperidine-1-oxyl-oxidized cellulose nanocrystals (TOCNC). The obtained composite particles were applied as stabilizers for acrylated epoxidized soybean oil (AESO) Pickering emulsion. The stability of the emulsion was characterized by determination of the rheological properties and microscopic morphology of the emulsion. The emulsion stabilized by composite particles showed better stability compared to case when TOCNC were used. The porous foam was obtained by heating a composite-particles-stabilized Pickering emulsion at 90 °C for 2 h. SEM (scanning electron microscopy) images showed that the prepared foam had uniformly distributed pores. In addition, the thermal conductivity of the foam was 0.33 W/m·k, which was a significant decrease compared to the 3.92 W/m·k of the TOCNC foam. The introduction of nisin and WPU can reduce the thermal conductivity of the foam, and the physically modified, TOCNC-stabilized Pickering emulsion provides an effective means to preparing bio-based porous materials.

## 1. Introduction

Porous foams are a class of materials with high porosities and large high specific surface areas, which are widely used in thermal insulation, energy storage, etc. At present, the production of most foams applied in the commercial field highly depends on petroleum resources, which limits their application. As fossil and petroleum resources become increasingly depleted, it is becoming a trend to use renewable resources and environmentally friendly production strategies to develop bio-based foams to replace traditional foam materials [1]. Various biomaterials have been reported for the preparation of bio-based foams. Hassan et al. [2] fabricated a biodegradable starch/cellulose composite foam with starch-containing cellulose fibers as a reinforcing agent and citric acid as a cross-linking agent. The composite foams can be applied as a biodegradable replacement for polystyrene foam. Qiu et al. [3] used AESO chemically grafted with flame retardants to produce a bio-based foam with desirable mechanical properties and flame retardancy. Luo et al. [4] reported a biodegradable poly (3-hydroxybutyrate-co-3-hydroxyvalerate) (PHBV)-based electromagnetic shielding foam by supercritical CO_2_, and constructed a green and economic avenue for the application of this PHBV foam in the field of electromagnetic shielding.

Emulsion is a dispersed system consisting of several immiscible liquids. Compared with traditional surfactant-stabilized emulsion, a Pickering emulsion generally has excellent stability due to nearly irreversible interfacial adsorption of stabilizers [5]. In addition, a Pickering emulsion also has environmentally friendly properties—e.g., a particle-stabilized emulsion can reduce the total amount of surfactant required. Based on the basic properties of a Pickering emulsion, it is an excellent template for preparing bio-based porous foam, and the pore structure of the foam can be adjusted by the stability of the emulsion and droplet size [6]. In a Pickering emulsion, particle stabilizers are particularly important for the stability of it and the properties of the prepared material. Some inorganic particles, such as nonmetallic oxide particles and magnetic particles, have been extensively studied in Pickering emulsions, but their common disadvantage is poor biocompatibility. In recent years, as a renewable material which has superior biocompatibility and biodegradability, nanocellulose, has attracted the interest of researchers.

Nanocellulose is a renewable and biodegradable nanomaterial that combines a large specific surface area, flexibility, low density and chemical inertness. Due to the presence of hydroxyl groups on the surface of nanocellulose, it can be physically modified or chemically modified with other polymers and nanomaterials by functional groups or grafting biomolecules [7]. In recent years, nanocellulose, which is carbon-neutral, non-toxic and sustainable, has attracted a lot of attention due to environmental concerns. At present, nanocellulose is widely applied in food packaging [8], thermal insulation materials [9,10], coatings [11], biomedicine [12], etc. Due to the surface properties, shape and inter-particle interactions of nanocellulose, it shows good self-assembly ability at the liquid interface and has been used to stabilize Pickering emulsions [13,14].The surface of nanocellulose has abundant OH groups, which gives it hydrophilicity overall, and the hydrophobic (200) β crystalline edges containing CH groups impart hydrophobicity [15]. The amphiphilicity of nanocellulose plays a crucial role in the stability of a Pickering emulsion. On the other hand, the stability of the emulsion generally affects the pore structure and properties of the prepared foam, including the thermal insulation and mechanical properties [7]. Zhang et al. [16] used aminated cellulose nanocrystals (CNC) to stabilize an oil-in-water (o/w), Pickering high internal phase emulsion (HIPE), and constructed a CNC aerogel with high porosity and low thermal conductivity by a simple Pickering emulsion template method. Capron et al. [17] reported a lightweight foam with porous structures which was obtained from freeze-drying the CNC stabilized o/w Pickering emulsion. Liu et al. [18] prepared a microwave absorbing foam by compounding the freeze-drying with the o/w Pickering emulsion gelation method, which emulsion was co-stabilized with CNF, carbon nanotubes (CNT) and Fe_3_O_4_ nanoparticles; and the prepared foams showed excellent thermal insulation properties compared to commercial polyvinyl alcohol and polyurethane foams. However, it still is a challenge to develop a strategy to prepare foams using nanocellulose-stabilized water-in-oil (w/o) Pickering emulsions.

In general, for a w/o Pickering emulsion, porous foams can be prepared through the polymerization and curing of the oil phase, and the function of foams can be adjusted by the choice of colloidal particles that are used as the stabilizer [19]. However, nanocellulose has high hydrophilicity and struggles to form a stable w/o Pickering emulsion by itself [20]. To improve the stability of nanocellulose in making Pickering emulsions, several attempts have been made, such as acetylation [21], quaternary surfactant adsorption [22] and organic-acid grafting [23].

Nisin is a cationic antimicrobial polypeptide with 34 amino acids which has been used as an antimicrobial agent in food. In recent years, nisin has mainly been combined with other polymeric matrices, such as polyethylene, nanocellulose, maize protein and starch, by physical and chemical methods [24]. Amino groups carried by nisin give it a positive charge. This allows it to be combined with other materials by electrostatic attraction, which gives it great potential for packaging material functionalization. Polyurethane (PU) is a copolymer containing repeated urethane groups composed of soft and hard chain segments. It has excellent mechanical properties, good compatibility and is easy to modify [25,26]. However, traditional PU could release a large amount of volatile organic compounds during its application. Therefore, environmentally friendly WPU is gaining more and more attention [27], and applying WPU to Pickering emulsions may be an interesting topic. Due to WPU’s deformable particle structure and functional groups, it is a potential emulsifier for the preparation of Pickering emulsions, and WPU particles, as soft colloids, can impart deformable interfaces to a Pickering emulsion [28].

With growing concern around environmental issues, incorporating plant-derived materials into current polymer systems to create new bio-based polymers or polymer foams is an attractive research topic. Vegetable oils are considered ideal raw materials for the preparation of sustainable foams because they are abundant and readily available. AESO is a soybean oil derivative consisting of triglyceride oil, which can be polymerized into high-molecular-weight and highly cross-linked thermoset polymers. In recent years, AESO has been used as a resin to form “green” composites or foam materials [29,30].

Herein, we submit a method for the preparation of bio-based porous foams using a physically modified, TOCNC-stabilized AESO Pickering emulsion. In our study, cationic nisin was immobilized on the surface of anionic TOCNC by electrostatic interaction, providing a basis for enhancing the interface stability of TOCNC as a Pickering-foam stabilizer [31]. On the basis of this, WPU was introduced to synergistically enhance the stability of TOCNC in the preparation of Pickering emulsions. The stability of Pickering emulsions was evaluated by microscopic observations and rheological tests. A Pickering emulsion was used as a template to prepare porous foam by thermal curing of AESO. Furthermore, the pore structure and thermal insulation properties of the foam were studied.

## 2. Materials and Methods

### 2.1. Materials

Nisin (≥1000 IU/mg) was purchased from Xinyinxiang Biological Engineering Co., Ltd. (Zhejiang, China). TOCNC were purchased from Zhejiang Jinjiahao Green Nanomaterials Co., Ltd. (Zhejiang, China). Hydrochloric acid and sodium hydroxide were purchased from Chengdu Kelong Chemical Reagent Factory (Sichuan, China). Dimethylolpropionic acid (DMPA), 1,4-butanediol (BDO), isophorone diisocyanate (IPDI), 4-hydroxyanisole, acetone, dibutyltin dilaurate, polybutylene glycols (PTMG), benzoyl peroxide (BPO), anhydrous acetone, triethylamine (TEA), epoxidized soybean oil (ESO), 4-methoxyphenol (MEHQ) and triphenylphosphine (TPP) were purchased from Aladdin Reagent Co., Ltd. (Shanghai, China). All chemical agents used in this research were analytical grade.

### 2.2. Preparation of WPU

WPU was synthesized in the laboratory using the following procedure. Firstly, a mixture of PTMG (30 g), acetone (60 g) and DMPA (2.68 g) were added into a four-necked flask equipped with a mechanical stirrer and thermometer. The reaction was carried out at 60 °C under argon for 20 min. Then IPDI (15.54 g) and dibutyltin dilaurate were added into the mixture, and the reaction was carried out at 80 °C for 4 h. After that, BDO (1.35 g) was added into flask, and the reaction was kept for 1 h. Subsequently, TEA (2.02 g) was added to neutralize the carboxylic groups in the prepolymer, and the temperature was reduced to 10 °C after 1 h. Finally, ultrapure water (103 g) was added into flask with a constant pressure funnel to obtain the WPU solution. The prepared WPU solution was stored at room temperature.

### 2.3. Preparation and Characterization of Particle Suspensions

Deionized water was added to the TOCNC aqueous dispersion and magnetically stirred for 30 min to prepare a TOCNC suspension (0.2 wt%). A TOCNC/nisin (TCN) suspension was prepared by adding a nisin solution (0.03 wt%) into a TOCNC suspension and magnetically stirring for 3 h. The TOCNC/nisin/WPU (TCNW) suspension was prepared by adding a WPU solution (0.01 wt%) into the TCN suspension and stirring for 1 h. All particle suspensions were prepared at pH = 7.

The transmission electron microscopy (TEM, HT7700, Hitachi, Japan) was used to observe the topographic characteristic of particles. The particle suspension was diluted to 0.005 wt% with deionized water; then a drop of particle suspension was deposited on a carbon-coated copper grid, followed by drying out overnight at 25 °C. The grid was observed using TEM at 100 kV.

The interfacial tension of the suspensions on the air–water interface was determined using a tensiometer (JK99F, Zhongchen Digital Technology Equipment Co., Ltd., Shanghai, China) at 25 °C and 60% relative humidity.

A high-speed blender (Unidrive 1000D, Ballrechten-Dottingen, Germany) was used to shear particle suspensions at 12,000 rpm for 2 min; then, the foaming heights were recorded at different times. The micromorphology of bubbles was studied by an optical microscope (Leica FSC, Leica Instruments Ltd., Weztlar, Germany).

### 2.4. Preparation and Characterization of Emulsions

The prepared particles were used as stabilizers for w/o Pickering emulsions. In our previous experiments, we used AESO Pickering emulsions as templates to prepare porous foams [19,32,33]. The effects of different particles and different oil–water ratios on the stability of emulsions were investigated by the characterization of centrifugal stability, rheological properties and viscosity of emulsions. In the experiments of using nanocellulose-stabilized AESO emulsions to the prepare foams, we found that the emulsions with water content of between 10–40% had better stability, so an oil–water ratio of 6:4 was chosen for preparing the Pickering emulsions in our study. Emulsions were prepared by mixing the oil phase with the particle suspension at a fixed oil-to-water ratio of 6:4 using a high-speed blender at 12,000 rpm for 3 min. Typically, a mixture of AESO, BPO (3 wt%), and anhydrous acetone was used as the oil phase. Among them, AESO was prepared according to the research method in [33].

The micromorphology of emulsions was studied using an optical microscope. The size distribution of emulsion droplets was determined by Nano Measurer 1.2 software.

The rheological properties of the emulsion were tested by a rotary rheometer (HAAKE MARS 40, Thermo Fisher, Karlsruhe, Germany). Oscillation amplitude frequency sweep with strain sweep from 0.1% to 100% was performed at 25 °C with an angle frequency of 1 rad/s to obtain the linear viscoelastic regions (LVR) of emulsions, and the limiting deformation value of the emulsion was given by the rheological system according to the variation in the storage modulus (G′) and the loss modulus (G″) within the LVR. The gap between two parallel plates (Φ = 35 mm) was set to 1 mm.

### 2.5. Preparation and Characterization of Foams

The foams were prepared by polymerization of AESO continuous phases in a Pickering emulsion system. The emulsions were transferred into a Teflon tube (diameter = 12.7 mm). A thermal polymerization reaction was carried out by heating the emulsions at 90 °C for 2 h. The prepared foams were dried at 50 °C overnight.

The micromorphology of foams was assessed by SEM (F16502, PHENOM, Eindhoven, Netherlands). Then, the area and size of pores in the range of 540 μm × 540 μm were measured by image J, and the specific surface of unit volume (*δ*) of foams was calculated according to Equation (1).
(1)δ=A/(L·W·d)
where *A* is the area of pores; *L* and *W* are the length and width of foams; and d is the average diameter of pores.

The thermal conductivity (*λ*) of foams was calculated according to Equation (2). The foam was cut into a sheet 12.7 mm in diameter and 1 mm thick; then, the density (*ρ*) of the sample was measured. The specific heat capacity (*Cp*) of each foam was measured by differential scanning calorimetry (DSC, 3500 Sirius, NETZSCH-Gerätebau GmbH, Selb, Germany). The initial and termination temperature of system were set to 20 and 40 °C, respectively, and the specific heat capacity of the foam was measured at 35 °C. The thermal diffusivity (*α*) of foams was measured by laser thermal conductivity meter (LFA467, NETZSCH-Gerätebau GmbH, Selb, Germany) at 35 °C.
(2)λ=α · Cp · ρ

## 3. Results and Discussion

### 3.1. Morphology and Properties of Stabilized Particles

To study the assembly structure of particles, TEM images of particles are shown in Figure 1. It can be seen that TOCNC were uniformly dispersed, short rod-shaped fibers. In order to improve the hydrophobicity of TOCNC, TCN particles were prepared by combining nisin with TOCNC. In the TEM images of TCN particles, TOCNC were entangled with each other and aggregated due to the electrostatic attraction with nisin. On the basis of TCN particles, WPU was introduced to prepare TCNW composite particles. In TCNW particles, WPU particles were adsorbed on the surface of aggregated TOCNC.

Generally, the efficacy of particles to stabilize the bubbles can be assessed by their ability to decrease interfacial tension. As shown in Figure 2a, the interfacial tension of the TOCNC suspension was 78.2 mN/m, which was obviously higher than those of other suspensions. The abundant hydroxyl groups on TOCNC provide hydrophilicity to make it have a less wettability, so it barely had the ability to reduce the interfacial tension. On the other hand, compared with TOCNC suspension, the interfacial tension of the TCN suspension decreased to 54.4 mN/m due to the cationic nisin being electrostatically adsorbed on the anionic TOCNC to improve the wettability of TOCNC at the gas–liquid interface [34]. The introduction of WPU made the interfacial tension of the TCNW suspension further decrease to 49.3 mN/m; the presence of hydrophobic and hydrophilic functional groups in WPU gives it a strong amphiphilic property at the gas–liquid interface. The suspension foaming heights at different times and foaming-effect pictures are shown in Figure 2b. The TOCNC suspension can only be seen to have had a very thin foam layer after shear foaming, TOCNC alone barely had foaming ability. The foaming effects of TCN and TCNW suspensions were obvious after shearing; they both had a thick foam layer, and the height of the foam layer decreased slowly with the time. In addition, the foaming height of the TCNW suspension decreased more slowly compared to that of the TCN suspension. It can be seen that both TCN and TCNW particles foam up better than TOCNC alone, and TCNW particles had the better foam-stabilization effect.

As shown in the optical microscope images of bubbles in Figure 3, the TCN-particles-stabilized bubbles were fewer, and their shape was irregular, whereas the TCNW-particles-stabilized bubbles were more, and they were small. A possible explanation for this discrepancy may be that the adsorption layer formed by TCN particles on the surface of a bubble is thin, meaning it cannot protect the bubble effectively, resulting in the bubbles being deformed under the influence of pressure. The addition of WPU or TCNW particles can form a denser particle shell at the gas–liquid interface to stabilize bubbles. This indicates that nisin and WPU have a synergistic effect on the stabilization of bubbles, and this combination could be used for stabilizing a w/o Pickering emulsion.

### 3.2. Stability of Pickering Emulsions

The stability and droplets size of the Pickering emulsion determine the pore structure and properties of the prepared foam. The micromorphology of emulsions was observed by optical microscope. As Figure 4 shows, the droplets of TOCNC emulsion were less, and the range of droplet size distribution was wide. In the TOCNC emulsion, it can be seen that the droplet size of the TOCNC emulsion was larger than those of TCN and TCNW emulsions, but all three emulsions were stable without creaming. We believe the large droplets have a strong tendency to aggregate with each other, which may result in the instability of the emulsion [35]. In a Pickering emulsion system, the size of the droplets is highly related to the amount of particle stabilizer on the surface of each droplet. In a TOCNC emulsion, due to the low interfacial wettability of TOCNC particles, the surface of droplets cannot be covered completely by particles, so the stability of the TOCNC emulsion was less than that of the other emulsions, thereby resulting in the formation of larger droplets in the emulsion. On the contrary, with the introduction of nisin and WPU, the interfacial wettability of TOCNC was improved, and the obtained composite particles could adsorb on the droplet’s surface to form a dense adsorption layer, thereby allowing the formation of smaller droplets. The TCNW emulsion had a narrower range of droplet size distribution, which provides a favorable condition for the preparation of porous foam.

The stability of emulsions can be determined by rheological testing. The modulus variation of emulsions within the LVR is shown in Figure 5. Typically, emulsions with a wide LVR have a larger limit deformation value, which indicates the emulsion structure has better stability [36]. Compared to the TOCNC emulsion, the limit deformation values of TCN and TCNW emulsions increased by 52% and 54%, respectively, indicating that they both had better stability. Within the LVR, three emulsions showed that G′ was lower than G″; the rheological behavior of emulsions was mainly viscous. The G″ of all emulsions were similar, though the G′ of TCN and TCNW emulsions had increased compared to that of TOCNC. A possible mechanism may be that G″ depends mainly on the continuous phase of emulsion, and the three emulsions have the same continuous phase of AESO; thus, they have similar G″ values. On the other hand, G′ is related to the droplet distribution in the dispersed phase, so uniform droplet distribution resulted in higher G′ for TCN and TCNW emulsions compared to the TOCNC one. In theory, the emulsion’s stability can be evaluated at the macroscopic level (bulk emulsion stability), mesoscopic level (droplet size and distribution) and microscopic level (interfacial shear rheology). From the microscopic point of view, the introduction of hydrophobic nisin and soft colloidal WPU facilitates the interface adsorption and elasticity of composite particles, benefiting the stability of the emulsion [36].

The stability of a Pickering emulsion is highly related to the adsorption behavior of the particles at the oil-water interface. In this regard, we have the following assumption. As Figure 6 shows, generally, due to the low interfacial wettability of TOCNC, resulting in its lesser adsorption at the oil–water interface, TOCNC particles remain individually dispersed on the droplet surface due to the electrostatic repulsion between the particles. In TCN particles, TOCNC with nisin patches on the surface bridge with adjacent fibers by electrostatic interactions, and the surface wettability of TOCNC is improved, so the adsorption of particles at the oil–water interface has increased, which in turn improves the stability of the emulsion. Unlike rigid particles, WPU as a soft colloid can give Pickering emulsions a deformable interface [28]. In TCNW particles, the introduction of WPU allows for a closer fiber alignment and forms a soft protective barrier on the surfaces of droplets, thereby the emulsion can be stabilized more efficiently. In a Pickering emulsion, we believe the thermal insulation performance of a foam is highly related to the stability of the emulsion. The stability of the emulsion tends to determine the pore structure of the prepared foam, thereby affecting the thermal insulation properties of the foam.

### 3.3. Micromorphology and Thermal Insulation Properties of Foams

After confirming that the physically modified TOCNC can stabilize w/o Pickering emulsion, porous foams were prepared using the Pickering emulsion as a template. The SEM images and pore size distributions of foams are shown in Figure 7. Typical pore structure for the prepared foams can be observed. However, compared to emulsion droplets, the pores were significantly larger. A probable hypothesis is that the loss of thermal stability and coalescence of the emulsion under the thermopolymerization process caused this [33]. In the SEM images of Figure 7, TOCNC foam has a less porous structure than other foams with a non-uniform pore size varying from 4.42 to 90.3 µm. The pores of TCN and TCNW foam were much larger, and TCN and TCNW particles could be observed on the pore wall. By calculating the specific surface of unit volume, the specific surface area of TOCNC foam was found to be 0.0235 µm^2^/µm^3^. On the other hand, due to the decrease in pore size and increase in the number of pores, the specific surface areas of TCN and TCNW foams increased to 0.0439 and 0.0612 µm^2^/µm^3^, respectively. In porous materials, a large specific surface area means that the foam has more pore walls, which means that the pores have a stronger capacity to absorb thermal radiation [37].

The thermal insulation performance of foam can be characterized by thermal conductivity. Figure 8a shows the thermal conductivity of foams prepared by different particle-stabilized Pickering emulsions. The thermal conductivity of the TOCNC foam was 3.92 W/m·K; and the thermal conductivities of TCN and TCNW foams were significantly lower, at 1.21 and 0.33 W/m·K, respectively. It can be seen that the thermal conductivity of foams decreased with the increase in the number of pores and the decrease in pore size. This phenomenon can be explained by the heat transfer mechanism of foam. The conduction path of heat in the foam is shown in Figure 8b. Generally, the heat conduction in porous material can be divided into gas conduction, solid conduction, radiation conduction and thermal convection within pores [38]. Convection is usually negligible for foams with pore diameters less than 3 mm. The overall thermal conductivity can be expressed as a superposition of the mechanisms, as follows in Equation (3) [39]:(3)$$\lambda_{t}=\lambda_{g}+\lambda_{s}+\lambda_{r}$$
where *λ_t_* is the overall thermal conductivity, *λ_g_* is the gas thermal conductivity, *λ_s_* is the solid thermal conductivity and *λ_r_* is the conductivity by radiation.

The above three thermal conductivities can be decreased by reducing the size of pore. Among them, gas conduction is the main mechanism of heat transfer in foam materials. Additionally, gas thermal conductivity can be calculated by Equation (4):(4)$$\lambda_{g}=\frac{\in_{VF}}{1+2\beta \cdot K_{n}}k_{g}$$
where *β* is the efficiency of energy transfer between the gas molecule and the pore wall; *k_g_* is the conductivity of the gas; *∈_VF_* is the void ratio of the foam; *K_n_* is the Knudsen number, which is the radio of mean free path of gas molecules (*l_mean_*) to the pore size (*φ_c_*), and can be calculated by Equation (5) [40]:(5)$$K_{n}=\frac{l_{mean}}{\varphi_{c}}$$

When the pore size is equal to or smaller than the average free range of the gas, *K_n_* increases, and the energy transfer through the gas molecules is reduced.

It can be known that Gas conduction is related to the pore size and the average free range of air molecules [9]. In porous foam, the gas thermal conductivity will decrease with the reduction of pore size due to the well-known Knudsen effect [41,42]. On the other hand, with an increase in pores, the foam has more pore walls and a larger specific surface area; the capacity of the foam to absorb, reflect and scatter has enhanced, thereby reducing the thermal conductivity of the foam [43,44].

## 4. Conclusions

In this study, bio-based foams were obtained by polymerization of an AESO Pickering emulsion which was stabilized by physically modifying TOCNC. After TOCNC was physically modified using WPU and nisin, the interfacial tension of particles was reduced from 78.2 to 49.3 mN/m, and foaming experiments confirmed that the introduction of WPU and nisin has a synergistic effect in enhancing the stability of gas–liquid interface. Rheological tests of the emulsions showed that the stability of composite-particles-prepared emulsions was improved. In the SEM images, the foams prepared by TCN and TCNW particles had a more uniform pore structure, and the specific surface area of TCNW foam increased to 0.0612 µm^2^/µm^3^. Due to the increases in pores and specific surface area, the thermal conductivity of the TCNW foam was reduced to 0.33 W/M·K compared to the TOCNC. In this study, WPU and nisin were used to physically modify TOCNC to improve the stability of emulsion, thereby obtaining the foam with a uniform pore structure, and the thermal conductivity of the foam was reduced. The strategy developed in this study is expected to be applied to bio-based porous materials in the packaging field.

## Figures and Tables

**Figure 1 polymers-14-05159-f001:**
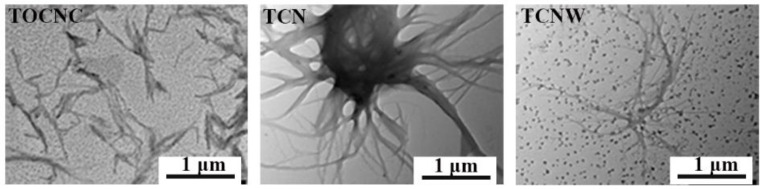
TEM images of TOCNC, TCN and TCNW particles.

**Figure 2 polymers-14-05159-f002:**
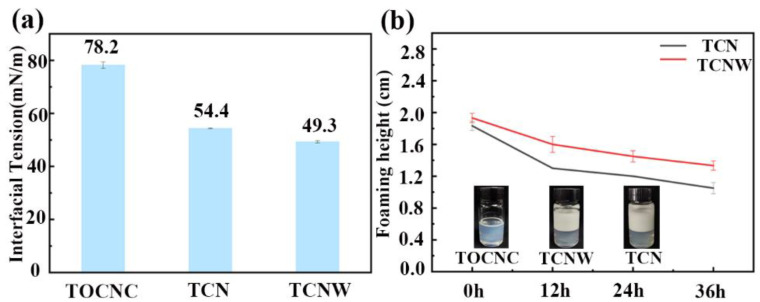
(**a**) Interfacial tensions of particle suspensions. (**b**) The suspension foaming heights at different times and foaming effect pictures.

**Figure 3 polymers-14-05159-f003:**
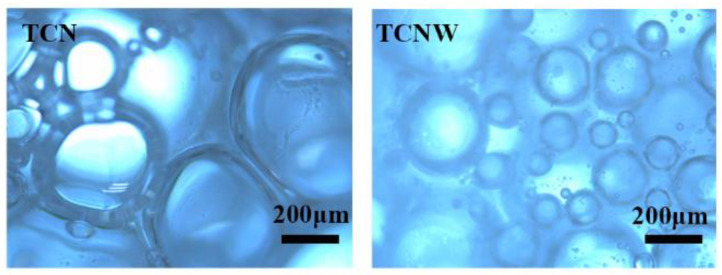
Optical micrographs of TCN- and TCNW-stabilized bubbles.

**Figure 4 polymers-14-05159-f004:**
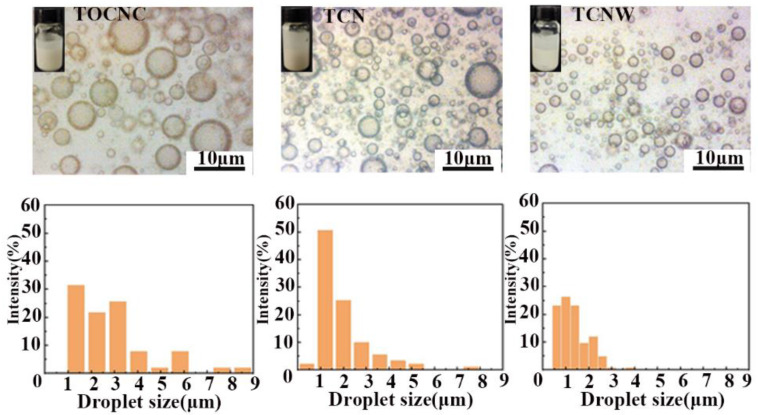
Optical micrographs and droplet size distribution of emulsions.

**Figure 5 polymers-14-05159-f005:**
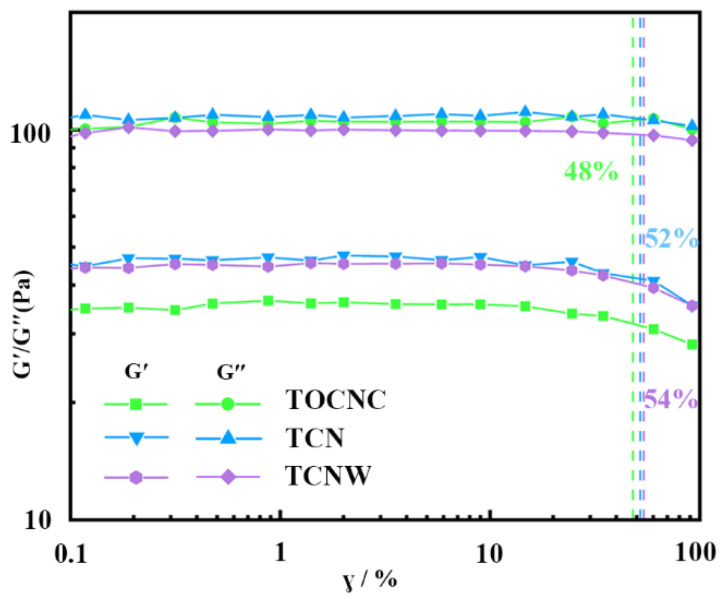
The changes of the storage moduli (G′) and loss moduli (G″) of emulsions.

**Figure 6 polymers-14-05159-f006:**
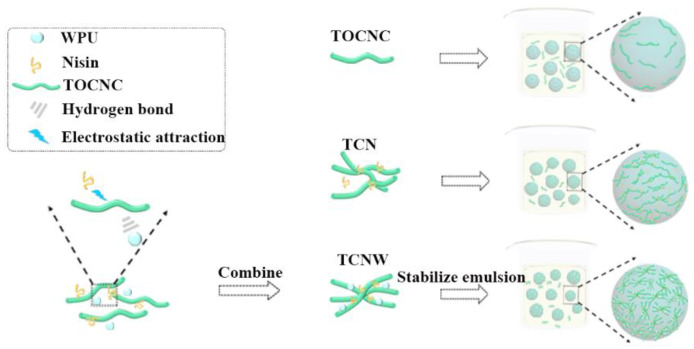
Adsorption of TOCNC, TCN and TCNW particles on emulsion droplets.

**Figure 7 polymers-14-05159-f007:**
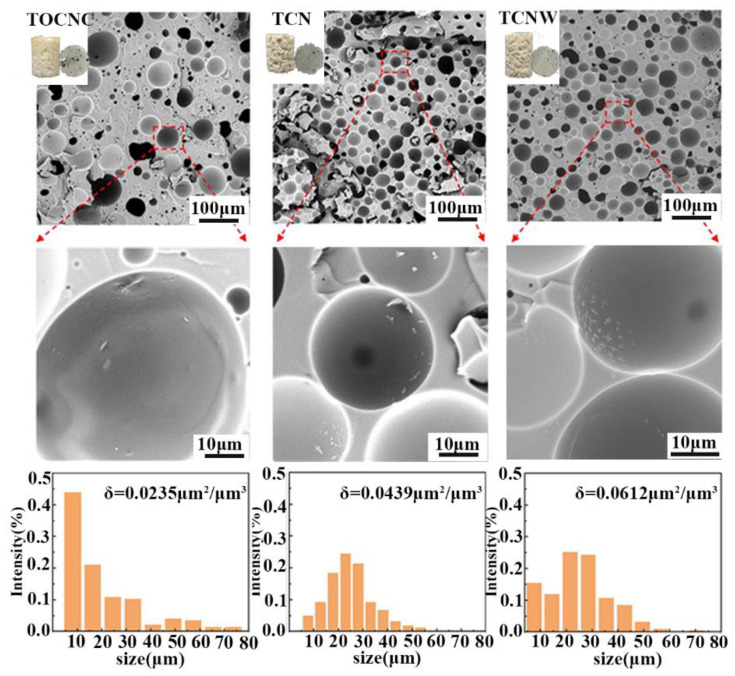
SEM images, specific surface of unit volumes and pore size distributions of foams.

**Figure 8 polymers-14-05159-f008:**
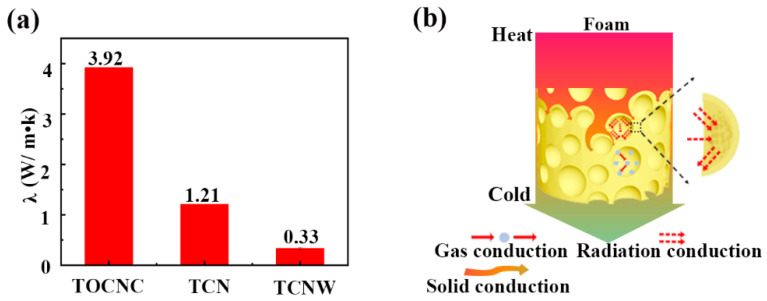
(**a**) Thermal conductivities of foams; (**b**) heat conduction in foams.

## Data Availability

Not applicable.

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
