# Peer review of "Preparation of Bio-Based Foams with a Uniform Pore Structure by Nanocellulose/Nisin/Waterborne-Polyurethane-Stabilized Pickering Emulsion"

_polymers, 2022, doi:10.3390/polym14235159_

Round 1
Reviewer 1 Report
In this study, bio-based porous foams were prepared and characterized i order to determine their rheological properties, microscopic morphology, thermal conductivity. The foams were synthesized by using Pickering emulsion as a template. Then, Nisin and waterborne polyurethane (WPU) were employed to physically modify 2,2,6,6-tetramethyl piperidine-1-oxyl-oxidized cellulose nanocrystals 11 (TOCNC). This was to construct the composite particle to stabilize epoxidized soybean oil (AESO) Pickering emulsion. The stability of emulsion was characterized by determining the rheological properties and observation of the microscopic morphology of emulsion.
General remarks
This article is interesting, but needs to be improved in terms of the language presentation. The question that comes to mind, is at what scale can these bio-based porous foams be produced and the cost-effectiveness? The big question is, how renewable are the resources and how friendly to the environment is the production strategy of the bio-based foams to replace traditional foam materials?It would have been interesting to determine how the the introduction of nisin and WPU influenced the porosity and mechanical properties of the bio-foams. Would it be possible to make comparisons of these foams in terms of thermal insulation performance, the thermal conductivity performance with the classical closed pore foams like extruded and expanded polystyrene? See some of the important but missing references.
@book{gibson_cellular_1997, title = {Cellular {Solids}: {Structure} and {Properties}}, isbn = {0 521 49560 1}, url = {https://doi.org/10.1017/CBO9781139878326},
publisher = {Cambridge University Press}, author = {Gibson, Lorna J. and Ashby, Michael F.}, year = {1997}, }
D. Baillis, R. Coquard Radiative and conductive thermal properties of foams Wiley-VCH Verlag GmbH & Co, KGaA (2008) p. 343–84
Erick Ogam, Z.E.A. Fellah, Géry Ogam, Identification of the mechanical moduli of closed-cell porous foams using transmitted acoustic waves in air and the transfer matrix method, Composite Structures, Volume 135, 2016, Pages 205-216, https://doi.org/10.1016/j.compstruct.2015.09.029.
How does one explain that only the storage moduli (G′) (Fig 5.) seems to give a good indication of the differences in composition?
Specific remarks
1. The typos in the title should be corrected "Preparation of Bio-based Porous Foams by Nanocellu- 2 lose/Nisin/ Wa-terborne Polyurethane Stabilized Pickering 3 Emulsion". It should be made clearer and as it is now, it is not clear. The title should also give an indication that characterization was done (rheological properties, microscopic morphology, thermal conductivity ).
2. Line 12, please correct the word "con-struct ", line 16, "stabi-lized " ...
3. "Line 309, in the phrase containing "by physically modified TOCNC" did you mean modifying.
Author Response
Dear editor,
Thank you for your kind letter of “Editor and Reviewer comments” concerning our manuscript “Preparation of Bio-Based Porous Foams by TOCNC/nisin/WPU Stabilized Pickering Emulsion (ID: polymers- 2007785). We revised the manuscript in accordance with the reviewers’ comments. Here below is our point-by-point response to all reviewer comments. A marked version of the revised manuscript has been submitted after addressing the comments below.
Peng Lu, Ph.D., (corresponding author)
Associate Professor
Institute of Light Industry and Food Engineering
Guangxi University
Nanning, 530004, CHINA
Tel: 0771-3237305
Email: lupeng-1984@163.com
Reviewer 1:
In this study, bio-based porous foams were prepared and characterized in order to determine their rheological properties, microscopic morphology, thermal conductivity. The foams were synthesized by using Pickering emulsion as a template. Then, Nisin and waterborne polyurethane (WPU) were employed to physically modify 2,2,6,6-tetramethyl piperidine-1-oxyl-oxidized cellulose nanocrystals (TOCNC). This was to construct the composite particle to stabilize epoxidized soybean oil (AESO) Pickering emulsion. The stability of emulsion was characterized by determining the rheological properties and observation of the microscopic morphology of emulsion.
- This article is interesting, but needs to be improved in terms of the language presentation. The question that comes to mind, is at what scale can these bio-based porous foams be produced and the cost-effectiveness? The big question is, how renewable are the resources and how friendly to the environment is the production strategy of the bio-based foams to replace traditional foam materials? It would have been interesting to determine how the introduction of nisin and WPU influenced the porosity and mechanical properties of the bio-foams. Would it be possible to make comparisons of these foams in terms of thermal insulation performance, the thermal conductivity performance with the classical closed pore foams like extruded and expanded polystyrene? See some of the important but missing references.
Response: We sincerely thank you for this comment. We have checked the whole manuscript and carefully corrected the grammar errors. In this study, we have developed a process for the preparation of bio-based foam materials by physically modified TOCNC stabilized AESO Pickering emulsion, and the porous foams can be obtained by free radical polymerization of AESO. The Pickering emulsion template method has the advantage of low energy consumption and is easy to product, which is conducive to large-scale production. Besides, as the stabilizer of emulsion, nanocellulose is an abundant, renewable nanomaterial, and it tends to require only a low addition amount in the preparation of Pickering emulsion, which is beneficial to reduce the production cost of foam. In the past few years, due to the growing environmental concerns, the replacement of petrochemical polymer with materials derived from nature is increasingly being considered. Acrylicized epoxidized soybean oil (AESO) which derives from vegetable oils is considered ideal raw materials for the preparation of sustainable foams because they are abundant and readily available, and AESO's polymers have been proven to be comparable to petroleum-based unsaturated polyester resins and have been used as resins to form "green" composites or foam materials. So the development of environmentally friendly, renewable, highly porous nanocomposite foam materials based on AESO is an attractive research topic.
Generally, it is hard to prepare stable w/o Pickering emulsion for TOCNC owing to its hydrophilicity. In this study, in order to improve the stability of TOCNC in the preparation of w/o emulsions, nisin was used to hydrophobic modify TOCNC to enhance the wettability of particles on oil-water interface. Besides, as a soft colloid, WPU can impart deformable interfaces to Pickering emulsion. Using nisin and WPU to synergistically enhance the stability of TOCNC in the preparation of emulsion, thereby obtaining the foam with uniform pore structure.
In recent years, research about the application of nanocellulose in Pickering emulsion is increasingly attractive. However, studies on the preparation of porous materials using nanocellulose stabilized w/o Pickering emulsion are still lacking in reports. The main contribution of this study is that we have developed a new strategy to use nanocellulose to prepare bio-based porous foam. In this study, we used nisin and WPU to synergistically enhance the stability of emulsions prepared by TOCNC to obtain the bio-based porous foam. Furthermore, in the measurement of thermal conductivity, we found that the addition of nisin and WPU could reduce the thermal conductivity of foam compared to TOCNC alone.
- How does one explain that only the storage moduli (G′) (Fig 5.) seems to give a good indication of the differences in composition?
Response: We sincerely thank you for this comment. In order to evaluate the stability of the emulsions, we characterized the rheological properties of emulsions. Within the LVR, three emulsions showed that G′ was less than G″, the rheological behavior of emulsions was mainly viscous. Besides, the G″ of all emulsions was similar, while the G′ of TCN and TCNW emulsions had increased compared to the TOCNC. A possible mechanism may be that G″ is depending mainly on the continuous phase of emulsion, while three emulsions have the same continuous phase of AESO, thus they have the similar G″. On the other hand, G′ is related to the droplet distribution of dispersed phase, uniform droplet distribution may result in higher G′ for TCN and TCNW emulsions compared to the TOCNC. In theory, the emulsion stability can be evaluated from macroscopic level (bulk emulsion stability), mesoscopic level (droplet size and distribution), and microscopic level (interfacial shear rheology). From the microscopic point of view, the introduction of nisin and soft colloidal WPU facilitates the interface adsorption and elasticity of composite particles, benefiting the stability of emulsion [1].
- The typos in the title should be corrected "Preparation of Bio-based Porous Foams by Nanocellulose/Nisin/ Waterborne Polyurethane Stabilized Pickering Emulsion". It should be made clearer and as it is now, it is not clear. The title should also give an indication that characterization was done (rheological properties, microscopic morphology, thermal conductivity)
Response: Thank you very much for your suggestion. We have checked it and corrected the error, and the title was changed into “Preparation of Bio-based Foams with Uniform Pore structure by Nanocellulose/Nisin/ Waterborne Polyurethane Stabilized Pickering Emulsion”.
- Line 12, please correct the word "con-struct ", line 16, "stabi-lized "
Response: Thank you for your careful review. We have checked the manuscript carefully and corrected the error.
- "Line 309, in the phrase containing "by physically modified TOCNC" did you mean modifying.
Response: Thank you for your careful review. We have changed the phrase containing "by physically modified TOCNC" into “physically modifying TOCNC”. At last, thank you very much for taking the time to review our manuscript and we highly appreciated for your valuable comments.
Reviewer 2:
The authors presented the possibilities of using Pickering emulsions for the stabilization of polyurethanes. Due to this combination, the thermal conductivity was significantly reduced due to the even distribution of pores.
- Literature very modest. There is a lot of material on the subject of bio-foams (paragraph 1 of the introduction). Please add literature
Response: Thank you very much for your suggestion. We have added the literatures on bio-based foam research in the first paragraph of introduction.
- The authors have not explained what Nisin is
Response: Thank you very much for your careful review. We have added description of nisin in the last paragraph on the page 4 of manuscript according to your suggestion.
- It is not entirely clear what kind of bio-foam was used for the research. This should be clarified at the very beginning of the article for research purposes.
Response: Thank you very much for your suggestion. In our study, acrylated epoxidized soybean oil (AESO) was used for the preparation of bio-foam. AESO is a soybean oil derivative consisting of triglyceride oil, which can be polymerized into high molecular weight and highly cross-linked thermoset polymers. In recently years, AESO has been used as resins to form "green" composites or foam materials. In addition, we have redescribed what kind of material was used to prepared the bio-foam on page 5 of manuscript according to your suggestion.
- The results of the tests with TEM were given in the test results and the method was not described in the Experiment section.
Response: Thank you very much for your careful review. We have added the description of TEM test method in the experimental section on page 7.
- In the research part of the article (items 2, 3, up to line 284) there are no references (references to literature).
Response: Thank you very much for your suggestion. We have added the references according to your suggestion.
- Werse 275, In porous materials, the specific surface area tends to have a significant effect on the thermal insulation properties of the materials.’’- Explain how surface area effects on the thermal insulation properties.
Response: Thank you very much for your careful review. From the radiation point of view, foam is a translucent medium, when thermal radiation is conducted through the pores, the radiation is absorbed, reflected and scattered by the pore walls [2]. The pore walls influence the attenuation of thermal radiation. With the decrease in cell size, the number of cells and consequently the cell walls increased, the foam has a higher specific surface area per unit volume, more radiation is absorbed and therefore, the radiation conduction is decreased [3].
- Werse 235 Theological or Reological?
Response: Thank you very much for your careful review. We have corrected “theological to rheological”.
- The methodology of the research described in the research results in lines 235-245 has not been described (and in conclusion, werse 311-312)。
Response:Thank you very much for your careful review. We have supplemented the rheological test method in the experimental section, and the rheological results of the emulsion were reanalyzed on page 12 of manuscript according to your suggestion. At last, thank you very much for taking the time to review our manuscript and we highly appreciated for your valuable comments.
Reviewer 3:
The manuscript is related to the development of a procedure to obtain bio-based porous materials built on the optimization of stability performance of Pickering emulsions. Generally, the investigation is of interest to the audience of Polymers.
Aside from just reporting the results however, it would be important to add some more clarification about the basic phenomena that stand behind the experimental outcomes. Besides, before further processing of the manuscript, some unclear statements and conclusions should be addressed in more depth and details, e.g.:
- line 35: “… stability due to irreversible adsorption of solid particles on the interface…“What does ‘irreversible’ mean here?
Response: Thank you for your careful review. Owing to the high energy of particle desorption from the interface, Pickering emulsions are often considered to be highly stable due to the nearly irreversible interfacial adsorption of particulate stabilizers [4].
- line 46: “…nanomaterial with high strength….” is an unclear statement.
Response: Thank you for your careful review. We have changed this description in the manuscript.
- lines 59-60, 195: What does ‘good hydrophilicity’ mean?
Response: Thank you very much for your careful review. We have corrected the description in the line 58-59 and line 185-186.
- line 145: How and why is ‘oil to water ratio 6:4’ chosen?
Response: We sincerely thank you for this comment. In our previous experiments, we have used AESO Pickering emulsions as template to prepare porous foams [5-7], the effect of different particles and different oil-water ratios on the stability of emulsions was investigated by the characterization of centrifugal stability, rheological properties and viscosity of emulsions. In the experiments of using nanocellulose stabilized AESO emulsions to prepare foam, we found that the emulsions with water phase between 10%-40% had a better stability, so we chose an oil-water ratio of 6:4 for our subsequent study.
- lines 193-194: Why is the interfacial tension so high (78.2 mN/m)?
Response: Thank you very much for your careful review. Due to the hydroxyl groups on the surface of the crystalline structure of TOCNC provides an overall hydrophilic character, the TOCNC particles have a less wettability, it barely has capability to reduce the interfacial tension, this causes the suspension to have a high interfacial tension.
- lines 213-215: “The bubbles stabilized by TCN particles are deformed due to squeezing each other, this makes it easier for the bubbles to burst and results that the height of foam layer of suspension decreases faster.” This is completely imprecise statement. What is actually the provisional mechanism of ‘deforming and ‘squeezing’? Do thin liquid films form in this case? What does ‘burst’ stand for?
Response: We sincerely thank you for this comment. We have redescribed it in the manuscript. In the microscopic observation of Pickering foam, the TCN particles stabilized bubbles were less and the shape was irregular, while the TCNW particles stabilized bubbles were more and the size was small. A possible explanation for this discrepancy may be that, the adsorption layer formed by TCN particles on the surface of bubble is thin, and which cannot protect the bubble effectively, resulting in the bubbles are deformed under the influence of pressure. While with the addition of WPU, TCNW particles can form a denser particle shell on gas-liquid interface to stabilize bubbles.
- lines 217-218: “TCNW particles can achieve better adsorption on gas-liquid interface to prevent the collapse of adjacent bubbles due to mutual squeezing, so the bubbles can remain for a longer time without disappearing…” Imprecise and unclear statement.
Response: Thank you very much for your careful review. We have redescribed it in page 11of manuscript.
- line 227: “…the droplets tend to aggregate with each other to form large droplets.” Do the droplets aggregate into complexes? How are these ‘large droplets’ formed?
Response: Thank you very much for your careful review. In the microscopic observation of the emulsion, the TOCNC emulsion had a non-uniform droplet distribution. In the TOCNC emulsion, due to the lack of sufficient interfacial wettability, the adsorption of TOCNC on oil-water interface is less, so that in droplets often show a tendency to polymerize during flow.
- lines 251-254: “In TCN emulsion, nisin can pull TOCNC closer to each other by electrostatic attraction, and allowing them to overlap each other and form a simple network structure on the oil-water interface, thereby improving the stability of Pickering emulsion.” This is completely imprecise reasoning. What do ‘óverlap each other’ and ‘simple network structure’ mean? What is the interplay mechanism of the various phenomena linked to ‘improving the stability of Pickering emulsion’ in this case? How are these related to the specific experimental conditions? Do the droplets aggregate into complexes? How are these ‘large droplets’ formed?
Response: We sincerely thank you for this comment. We have corrected this description according to your suggestion. Generally, due to the less interfacial wettability of TOCNC, resulting in its less adsorption on oil-water interface, and TOCNC particles remain individually dispersed on the droplet surface due to the electrostatic repulsion between the particles. In TCN particles, TOCNCs with nisin patches on the surface bridge with adjacent fibers by electrostatic interaction, and the surface wettability of TOCNCs is improved, so the adsorption of particles on the oil-water interface has increased, which in turn improves the stability of the emulsion. Unlike rigid particles, WPU as a soft colloid can give Pickering emulsions a deformable interface. The introduction of WPU allows for a closer fiber alignment and form a soft protective barrier on the surface of droplets, so the emulsion has a uniform droplet distribution and better stability. At last, thank you very much for taking the time to review our manuscript and we highly appreciated for your valuable comments.
Reviewer 4:
- English text should be improved. Abstract, Introduction, Experimental and Conclusions Sections should be rewritten.
Abstract
“Bio-based porous materials can reduce energy consumption and environmental impact, and have a promising application in packaging. In this study, the bio-based porous foam was prepared by using Pickering emulsion as a template. Nisin and waterborne polyurethane (WPU) was used to physically modify 2,2,6,6-tetramethyl piperidine-1-oxyl-oxidized cellulose nanocrystals (TOCNC) to con-struct the composite particle to stabilize epoxidized soybean oil (AESO) Pickering emulsion. The stability of emulsion was characterized according to the rheological properties and microscopic morphology of emulsion. The emulsion stabilized by composite particles showed better stability compared to TOCNC alone.”
Should be replaced by
“Bio-based porous materials can reduce energy consumption and environmental impact, and they have a possible application as packaging materials. In this study, the bio-based porous foam was prepared by using Pickering emulsion as a template. Nisin and waterborne polyurethane (WPU) was used for physical modification of 2,2,6,6-tetramethyl piperidine-1-oxyl-oxidized cellulose nanocrystals (TOCNC). The obtained composite particles are applied as stabilizers for epoxidized soybean oil (AESO) Pickering emulsion. The stability of emulsion was characterized by determination of rheological properties and microscopic morphology of emulsion. The emulsion stabilized by composite particles show better stability compared to case when TOCNC is used.”
Response: Thank you very much for your careful review and suggestions on our manuscript. We have checked the whole manuscript and tried our best to improve our English writing skills. Besides, we have rewritten Abstract, Introduction, Experimental and Conclusions Sections according to your suggestion.
- Please define more precise the aim of this work in the end of Introduction section.
Response: Thank you for your suggestion. We have redescribed the purpose of the study in the end of Introduction section.
- Page 3, row 115 “TPDI” should be corrected to “IPDI”
Response: Thank you for your suggestion. We have checked it carefully and corrected errors.
- page 3, row 99, “Experiment” should be substituted with “Materials and methods”
Response: Thank you for your suggestion. We have checked it carefully and corrected errors.
- page 9, row 307, “Conclusion” should be replaced by “Conclusions”
Response: Thank you for your suggestion. We have checked the manuscript and changed it.
- Please add DOIs of the references.
Response: Thank you for your careful review. We have added the DOIs in the references. At last, thank you very much for taking the time to review our manuscript and we highly appreciated for your valuable comments.
References
- Jia Y, Kong L, Zhang B, Fu X, Huang Q (2022) Fabrication and characterization of Pickering high internal phase emulsions stabilized by debranched starch-capric acid complex nanoparticles. International Journal of Biological Macromolecules 207:791-800. doi: https://doi.org/10.1016/j.ijbiomac.2022.03.142
- Baillis D, Coquard R (2008) Radiative and Conductive Thermal Properties of Foams. In: Cellular and Porous Materials. pp 343-384. doi: https://doi.org/10.1002/9783527621408.ch11
- Hasanzadeh R, Fathi S, Azdast T, Rostami M (2020) Theoretical Investigation and Optimization of Radiation Thermal Conduction of Thermal-Insulation Polyolefin Foams %J Iranian Journal of Materials Science & Engineering. 17 (2):58-65. doi: https://doi.org/10.22068/ijmse.17.2.58
- Jiang H, Sheng YF, Ngai T (2020) Pickering emulsions: Versatility of colloidal particles and recent applications. Current Opinion in Colloid & Interface Science 49:1-15. doi:10.1016/j.cocis.2020.04.010
- Lu P, Zhao H, Zhang M, Bi X, Ge X, Wu M (2022) Thermal insulation and antibacterial foam templated from bagasse nanocellulose /nisin complex stabilized Pickering emulsion. Colloids Surf B Biointerfaces 220:112881. doi: https://doi.org/10.1016/j.colsurfb.2022.112881
- Lu P, Guo M, Yang Y, Wu M (2018) Nanocellulose Stabilized Pickering Emulsion Templating for Thermosetting AESO Nanocomposite Foams. Polymers 10 (10). doi: https://doi.org/10.3390/polym10101111
- Tian X, Ge X, Guo M, Ma J, Meng Z, Lu P (2021) An antimicrobial bio-based polymer foam from ZnO-stabilised pickering emulsion templated polymerisation. Journal of Materials Science 56 (2):1643-1657. doi: https://doi.org/10.1007/s10853-020-05354-3
Reviewer 2 Report
Review article ,,Preparation of Bio-based Porous Foams by Nanocellulose/Nisin/ Wa-terborne Polyurethane Stabilized Pickering Emulsion’’
The authors presented the possibilities of using Pickering emulsions for the stabilization of polyurethanes. Due to this combination, the thermal conductivity was significantly reduced due to the even distribution of pores.
1. Literature very modest. There is a lot of material on the subject of bio-foams (paragraph 1 of the introduction). Please add literature.
2. The authors have not explained what Nisin is.
3. It is not entirely clear what kind of bio-foam was used for the research. This should be clarified at the very beginning of the article for research purposes.
4. The results of the tests with TEM were given in the test results and the method was not described in the Experiment section.
5. In the research part of the article (items 2, 3, up to line 284) there are no references (references to literature).
6. Werse 275 ,, In porous materials, the specific surface area tends to have a significant effect on the thermal insulation properties of the materials.’’- Explain how surface area effects on the thermal insulation properties.
7. Werse 235 Theological or Reological?
8. The methodology of the research described in the research results in lines 235-245 has not been described (and in conclusion, werse 311-312).

Author Response

(The authors gave the same response as above.)

Reviewer 3 Report
The manuscript is related to the development of a procedure to obtain bio-based porous materials built on the optimization of stability performance of Pickering emulsions. Generally, the investigation is of interest to the audience of Polymers.
Aside from just reporting the results however, it would be important to add some more clarification about the basic phenomena that stand behind the experimental outcomes. Besides, before further processing of the manuscript, some unclear statements and conclusions should be addressed in more depth and details, e.g.:
1. line 35: “… stability due to irreversible adsorption of solid particles on the interface…“
What does ‘irreversible’ mean here?
2. line 46: “…nanomaterial with high strength….” is an unclear statement.
3. lines 59-60, 195: What does ‘good hydrophilicity’ mean?
4. line 145: How and why is ‘oil to water ratio 6:4’ chosen?
5. lines 193-194: Why is the interfacial tension so high (78.2 mN/m)?
6. lines 213-215: “The bubbles stabilized by TCN particles are deformed due to squeezing each other, this makes it easier for the bubbles to burst and results that the height of foam layer of suspension decreases faster.”
This is completely imprecise statement. What is actually the provisional mechanism of ‘deforming and ‘squeezing’? Do thin liquid films form in this case? What does ‘burst’ stand for?
7. lines 217-218: “TCNW particles can achieve better adsorption on gas-liquid interface to prevent the 217 collapse of adjacent bubbles due to mutual squeezing, so the bubbles can remain for a longer time without disappearing…”
Imprecise and unclear statement.
8. line 227: “…the droplets tend to aggregate with each other to form large droplets.”
Do the droplets aggregate into complexes? How are these ‘large droplets’ formed?
9. lines 251-254: “In TCN emulsion, nisin can pull TOCNC closer to each other by electrostatic attraction, and allowing them to overlap each other and form a simple network structure on the oil-water interface, thereby improving the stability of Pickering emulsion.”
This is completely imprecise reasoning. What do ‘óverlap each other’ and ‘simple network structure’ mean? What is the interplay mechanism of the various phenomena linked to ‘improving the stability of Pickering emulsion’ in this case? How are these related to the specific experimental conditions?
Author Response

(The authors gave the same response as above.)

Reviewer 4 Report
Review of the Manuscript Polymers-2007785
1. English text should be improved. Abstract, Introduction, Experimental and Conclusions Sections should be rewritten.
Abstract
“Bio-based porous materials can reduce energy consumption and environmental impact, 8 and have a promising application in packaging. In this study, the bio-based porous foam was pre-9 pared by using Pickering emulsion as a template. Nisin and waterborne polyurethane (WPU) was used to physically modify 2,2,6,6-tetramethyl piperidine-1-oxyl-oxidized cellulose nanocrystals (TOCNC) to con-struct the composite particle to stabilize epoxidized soybean oil (AESO) Pickering emulsion. The stability of emulsion was characterized according to the rheological properties and microscopic morphology of emulsion. The emulsion stabilized by composite particles showed better stability compared to TOCNC alone.”
Should be replaced by
“Bio-based porous materials can reduce energy consumption and environmental impact, and they have a possible application as packaging materials. In this study, the bio-based porous foam was prepared by using Pickering emulsion as a template. Nisin and waterborne polyurethane (WPU) was used for physical modification of 2,2,6,6-tetramethyl piperidine-1-oxyl-oxidized cellulose nanocrystals (TOCNC). The obtained composite particles are applied as stabilizers for epoxidized soybean oil (AESO) Pickering emulsion. The stability of emulsion was characterized by determination of rheological properties and microscopic morphology of emulsion. The emulsion stabilized by composite particles show better stability compared to case when TOCNC is used.”
2. Please define more precise the aim of this work in the end of Introduction section.
3. Page 3, row 115
“TPDI” should be corrected to “IPDI”
4. page 3, row 99, “Experiment” should be substituted with “Materials and methods”
5. page 9, row 307, “Conclusion” should be replaced by “Conclusions”
6. Please add DOIs of the references.

Author Response

(The authors gave the same response as above.)

Round 2
Reviewer 1 Report
The authors have revised the paper adequately in order to improve the paper.
Author Response
Dear Dr. Murphy Wang,
Thank you for your kind letter of “Editor and Reviewer comments” concerning our manuscript “Preparation of Bio-based Foams with Uniform Pore Structure by Nanocellulose/Nisin/ Waterborne Polyurethane Stabilized Pickering Emulsion (ID: polymers- 2007785)”. We revised the manuscript in accordance with the reviewers’ comments. Here below is our response to reviewer comments. A marked version of the revised manuscript has been submitted after addressing the comments below.
Peng Lu, Ph.D., (corresponding author)
Associate Professor
Institute of Light Industry and Food Engineering
Guangxi University
Nanning, 530004, CHINA
Tel: 0771-3237305
Email: lupeng-1984@163.com
Reviewer 1:
The authors have revised the paper adequately in order to improve the paper.
Response: Thank you very much for taking the time to review our manuscript and response, and we highly appreciated for your valuable comments which are very helpful in improving our manuscript.
Reviewer 3:
The replies to my initial comments are not well-specified and not included in the manuscript text, namely:
- Re: 4. line 145: How and why is ‘oil to water ratio 6:4’ chosen? The reply should be added in the manuscript.
Response: We sincerely thank you for this comment. We have added it on the page 7 of manuscript according to your suggestion.
- Re: 8. line 227: “…the droplets tend to aggregate with each other to form large droplets.” Do the droplets aggregate into complexes? How are these ‘large droplets’ formed?
Reply: "....droplets often show a tendency to polymerize during flow." This is an incorrect statement.
Response: Thank you very much for pointing out the shortcomings of our response. We have rewritten the reply to make it clearer. In emulsions, the droplets cannot aggregate into complexes, because there no creaming was observed during the emulsion preparation and storage. Moreover, no complexes were observed from the view of microscope images. From the microscope images, we can see that the droplet size of TOCNC emulsion was larger than TCN and TCNW emulsion, but all three emulsions were stable without creaming, so we claimed that droplets are not aggregate into complexes. Here, we believe the large droplets have a high tendency to aggregate with each other, which may result in the instability of emulsion [1]. In Pickering emulsion system, the size of droplet is highly related to the amount of particle stabilizer on the surface of individual droplet. In TOCNC emulsion, due to the low interfacial wettability of TOCNC particles, the surface of droplets cannot be covered completely by particles, so the stability of TOCNC emulsion was lower than other emulsions, thereby resulting in the formation of larger droplets in emulsion. On the contrary, with the introduction of nisin and WPU, the interfacial wettability of TOCNC was improved, and the obtained composite particles can adsorb on the droplet surface to form a dense adsorption layer, thus allowing the formation of smaller droplets. Besides, we have also redescribed it on the page 11 of manuscript. At last, thank you very much for taking the time to review our manuscript and we highly appreciated for your valuable comments.
References
- Tcholakova S, Denkov ND, Sidzhakova D, Ivanov IB, Campbell B (2003) Interrelation between Drop Size and Protein Adsorption at Various Emulsification Conditions. Langmuir 19 (14):5640-5649. doi: https://doi.org/10.1021/la034411f
Reviewer 3 Report
The replies to my initial comments are not well-specified and not included in the manuscript text, namely:
Re: 4. line 145: How and why is ‘oil to water ratio 6:4’ chosen?
The reply should be added in the manuscript.
Re: 8. line 227: “…the droplets tend to aggregate with each other to form large droplets.” Do the droplets aggregate into complexes? How are these ‘large droplets’ formed?
Reply: "....droplets often show a tendency to polymerize during flow."
This is an incorrect statement.
Author Response

(The authors gave the same response as above.)
